# Pulmonary Hypertension Secondary to Fungal Infections: Underexplored Pathological Links

**DOI:** 10.3390/idr17040084

**Published:** 2025-07-12

**Authors:** Andrea Jazel Rodríguez-Herrera, Sabrina Setembre Batah, Maria Júlia Faci do Marco, Carlos Mario González-Zambrano, Luciane Alarcão Dias-Melicio, Alexandre Todorovic Fabro

**Affiliations:** 1Department of Pathology and Forensic Medicine, Ribeirão Preto Medical School (FMRP), University of São Paulo (USP), Ribeirão Preto 14049-900, SP, Brazil; ajrh9993@gmail.com (A.J.R.-H.); sabrina7batah@gmail.com (S.S.B.); mariajuliadomarco@usp.br (M.J.F.d.M.); 2Laboratory of Immunopathology and Infectious Agents (LIAI), Experimental Research Unity (UNIPEX), Sector 5, Botucatu Medical School, São Paulo State University (UNESP), Botucatu 18618-687, SP, Brazil; carlos.gonzalez-zambrano@unesp.br (C.M.G.-Z.); dias.melicio@unesp.br (L.A.D.-M.); 3Department of Pathology, Botucatu Medical School, São Paulo State University (UNESP), Botucatu 18618-687, SP, Brazil

**Keywords:** pulmonary hypertension, fungal infection, pathology, vascular remodeling

## Abstract

Background/Objective: Pulmonary fungal infections are a significant diagnostic challenge, primarily affecting immunocompromised individuals, such as those with HIV, cancer, or organ transplants, and they often lead to substantial morbidity and mortality if untreated. These infections trigger acute inflammatory and immune responses, which may progress to chronic inflammation. This process involves myofibroblast recruitment, the deposition of extracellular matrix, and vascular remodeling, ultimately contributing to pulmonary hypertension. Despite its clinical relevance, pulmonary hypertension secondary to fungal infections remains under-recognized in practice and poorly studied in research. Results/Conclusion: This narrative mini-review explores three key mechanisms underlying vascular remodeling in this context: (1) endothelial injury caused by fungal emboli or autoimmune reactions, (2) direct vascular remodeling during chronic infection driven by inflammation and fibrosis, and (3) distant vascular remodeling post-infection, as seen in granulomatous diseases like paracoccidioidomycosis. Further research and clinical screening for pulmonary hypertension in fungal infections are crucial to improving patient outcomes.

## 1. Introduction

Pulmonary fungal infections represent a significant diagnostic challenge [1], contributing to varying degrees of morbidity and mortality [2]. These infections primarily affect immunocompromised individuals, including those with human immunodeficiency virus (HIV), cancer, and organ transplantations or undergoing immunosuppressive therapy. If left untreated, these infections are associated with high rates of morbidity and mortality [2,3].

Fungal infections trigger acute inflammatory and immune responses that can progress to chronic inflammation. This process involves the recruitment of myofibroblasts and the deposition of extracellular matrix [4,5], eventually leading to vascular remodeling and pulmonary hypertension. In addition to immune-mediated vascular remodeling, clinical conditions such as prolonged mechanical ventilation, especially in patients with ARDS and concurrent fungal infections, may also contribute to the development of pulmonary hypertension through sustained endothelial injury and dysregulated inflammation [6]. While the individual and synergistic effects of these factors remain insufficiently characterized, their potential contribution underscores the importance of further investigation in this patient population.

Despite the potential severity of these complications, the assessment of pulmonary hypertension in patients with fungal infections is not commonly performed in clinical practice or research protocols [7]. This may be because the association has been largely overlooked and remains under-recognized.

We hypothesize that pulmonary fungal infections may play a direct role in the development of PH through persistent vascular inflammation, immune dysregulation, and structural remodeling of the pulmonary vasculature.

The pathophysiological mechanisms linking fungal infections to pulmonary hypertension remain poorly understood but are likely multifactorial, involving persistent vascular inflammation, immune dysregulation, and structural remodeling of the pulmonary vasculature. This narrative mini-review explores the three main mechanisms of vascular remodeling associated with fungal infections (Figure 1 and Figure 2), arguing that these infections are an under-recognized contributor to pulmonary hypertension. By consolidating the current evidence, we aim to highlight a critical gap in pulmonary medicine and encourage future research into this emerging intersection of infectious and vascular pathology.

## 2. Vascular Remodeling Post Endothelial Injury

Fungal emboli and secondary autoimmune reactions triggered by fungal infections are significant causes of endothelial injury. These mechanisms are most commonly described in aspergillosis [8,9], where clinical reports have documented fatal cases of *Aspergillus* endocarditis, particularly in cardiac surgery patients, often presenting with massive embolic events such as pulmonary embolism, cerebral infarctions, or peripheral arterial occlusions [8]. In a pediatric case, Miranda et al. reported a 4-year-old boy with a prosthetic pulmonary valve who developed septic pulmonary embolism due to *Aspergillus fumigatus* endocarditis, which progressed to secondary pulmonary hypertension and fatal pulmonary hypertensive crisis despite surgical debridement and intensive antifungal therapy [9].

Upon reaching the pulmonary vessels, fungal elements or autoantibodies induce endothelial damage, triggering an inflammatory cascade that promotes thrombosis, intimal thickening, and alterations in the medial and adventitial layers [5]. A structure involved in the formation of thrombus is NETs (neutrophil extracellular traps); these are primarily released by activated neutrophils and play a role in the pathogenesis of various autoimmune and inflammatory diseases, as well as endemic and systemic mycoses, and may have a determining role in pulmonary hypertension through the activation of platelets and endothelial cells [2,10,11,12,13]. This role is further supported by histopathological findings in lung tissue from patients with idiopathic pulmonary arterial hypertension (iPAH). Aldabbous et al. [14] identified NET components—such as DNA, myeloperoxidase (MPO), and citrullinated histone H3—within occlusive plexiform lesions and perivascular infiltrates. PAD4-positive neutrophils were also observed in the walls of remodeled vessels, reinforcing the involvement of NETs in vascular inflammation and remodeling in human PAH.

The endothelial damage induced by NETs, along with their ability to capture prothrombotic plasma proteins, highlights their central role in the development of immunothrombosis. NET components, particularly histones, exhibit cytotoxic effects on endothelial cells, leading to their activation and death in murine sepsis models [15]. Moreover, experimental evidence from in vitro and animal studies has shown that NETs contribute significantly to the formation of deep vein thrombosis (DVT) and the development of thrombus by activating FXII, inhibiting plasma tissue factor pathway inhibitor (TFPI), stimulating platelet activation, and capturing von Willebrand factor (VWF) [16,17,18,19]. Additionally, extracellular histones in NETs impair endothelial anticoagulant and anti-inflammatory pathways by reducing thrombomodulin (TM) cofactor activity, thereby decreasing the generation of activated protein C (APC) [20]. Furthermore, COVID-19-associated pulmonary aspergillosis (CAPA) has been linked to endothelial damage and thromboembolic events, factors that may promote the development of pulmonary hypertension. A recent case report reported pulmonary hypertension secondary to pulmonary embolism in a post-COVID patient with CAPA [21]. Although the evidence is still limited, this suggests a possible contribution of CAPA to pulmonary vascular damage.

All these structural changes significantly impair vascular function and contribute to the progression of pulmonary hypertension.

## 3. Direct Vascular Remodeling During Infection

Fungal infections induce acute local inflammation, which, when unresolved, becomes chronic. This process results in tissue softening due to inflammatory cell infiltration and hardening caused by the deposition of extracellular matrix and fibrosis. Together, these changes drive direct vascular remodeling during infection [4].

This phenomenon has been well-documented in aspergillosis in some case reports [22,23]. Remodeling may occur even in the presence of robust Th1 and Th17 immune responses in a model murine of *Aspergillus fumigatus* infection [24]. Conversely, CD4+ T-cell-independent, IL-10-dependent pathways have also been implicated in pulmonary arterial remodeling following repeated conidial exposure [24]. Experimental studies further highlight the chronicity of pulmonary hypertension and its associated morphological changes [25].

Swain et al. also showed that CD4+ T cells and IFN-gamma are required for the development of *Pneumocystis*-associated pulmonary hypertension, once wild-type mice developed pulmonary hypertension but IFN-gamma knockout mice did not [26]. In addition to dependence on IFN-gamma, it was found that when CD4 cells were continuously depleted, and infection was limited by treatment, pulmonary hypertension did not occur, confirming that CD4+ T cells are required for the development of pulmonary hypertension, and that vascular remodeling accompanied pulmonary hypertension, and it is associated with perivascular fibrosis [26,27].

Many fungal pathogens, such as *Candida* spp. [28,29] *Aspergillus fumigatus* [30], *Histoplasma capsulatum* [31], *Phialophora verrucosa* [32], *Cryptococcus neoformans* [33], *Trichophyton rubrum* [34], *Paracoccidioides brasiliensis* [35], and *Scedosporium apiospermum* [36], have been described as inducers of NETs’ release [37] as demonstrated in vitro through the exposure of human neutrophils to these fungal pathogens. A clinical study further identified NETs in the tegumentary lesions of patients with chronic paracoccidioidomycosis [38]. NETs have also been shown to contribute to neutrophil-driven inflammation by activating epithelial cells to secrete chemokines, including CXCL-1 and 8 [39,40].

Since NETs have been identified in a wide range of respiratory conditions—including cystic fibrosis [41,42], chronic obstructive pulmonary disease (COPD) [43,44], bronchiectasis [45,46,47,48], asthma [39,49], pneumonia [50,51], acute respiratory distress syndrome (ARDS) [52,53,54,55,56,57], COVID-19 [58,59,60,61,62,63,64,65,66,67], lung cancer [68,69,70,71,72], and interstitial lung disease (ILD) [73,74,75,76,77]—and are associated with the pathogenesis of these diseases due to different mechanisms [40,52], mainly through neutrophil-driven inflammation, we hypothesize that pulmonary hypertension could also result from vascular remodeling in damaged lung tissue triggered by fungal infections and NET-mediated injury.

## 4. Distant Vascular Remodeling Post-Infection

Granulomatous infections, such as those caused by *Paracoccidioides* species, have been associated with pulmonary hypertension and cor pulmonale in a substantial proportion of patients (Table 1). Yepez et al. [78] provided detailed descriptions of five autopsy cases of patients with paracoccidioidomycosis (PCM). Pulmonary hypertension was identified in two of the cases, and all five presented with cor pulmonale. Pulmonary histopathological findings included intravascular thrombosis, granulomas, fibrosis, and emphysema.

Machado Filho et al. [79] reported late-stage pulmonary hypertension in 23% of post-PCM cases. Campos et al. [80] demonstrated cor pulmonale in 24% of late-stage post-PCM patients (n = 58), underscoring the progressive nature of vascular remodeling. Notably, five of these patients died, highlighting the potential severity of cardiopulmonary involvement and the importance of routinely assessing pulmonary hypertension in chronic PCM cases.

Beyond the traditional *P. brasiliensis*, Gaspar et al. [81] reported pulmonary hypertension in a patient with *P. lutzii.* In this case report, the patient continued to present persistent respiratory symptoms after antifungal treatment, and pulmonary arterial hypertension was confirmed by computed tomography and echocardiography.

Recently, Batah et al. [82] highlighted the role of adventitial remodeling in the development of pulmonary hypertension, supported by both human biopsy and experimental data. The study investigates pulmonary hypertension induced by pulmonary paracoccidioidomycosis, a fungal infection caused by *Paracoccidioides brasiliensis*, aiming to determine whether PCM triggers long-term vascular remodeling leading to PH, even after clinical cure. While PCM is treatable, a significant subset of patients previously considered cured has been observed to develop progressive pulmonary vascular changes culminating in PH. This study provides novel evidence that PCM-induced PH is not merely a consequence of chronic lung damage but may represent an independent pathophysiological process.

## 5. Underlying Mechanisms of PCM-Induced Pulmonary Hypertension

An experimental model using Wistar rats was developed, where the intrapulmonary inoculation of *Paracoccidioides* yeast resulted in well-formed granulomas localized exclusively in the lung [82]. These animals exhibited a significant thickening of the adventitial collagen layer in pulmonary vessels, an increased deposition of extracellular matrix, and a marked elevation in right ventricular systolic pressure (RVSP), indicative of the development of PH. A notable reduction in small precapillary vessels was also observed, a common feature of precapillary PH. The activation of adventitial myofibroblasts was confirmed through immunohistochemistry for α-smooth muscle actin (α-SMA) and electron microscopy, which identified fibronexus structures, supporting the hypothesis that PCM-induced vascular remodeling is mediated by persistent inflammatory and fibrotic mechanisms.

A histopathological analysis of lung biopsies from 15 human PCM patients, diagnosed between 2007 and 2017, revealed findings consistent with the animal model [82]. Significant deposition of adventitial collagen fiber was observed, particularly in small pulmonary vessels (<50 μm), independent of the granuloma’s proximity, suggesting a sustained fibrotic response rather than direct fungal invasion. This pattern of vascular remodeling resembles that seen in fibrosing mediastinitis secondary to histoplasmosis and idiopathic pulmonary fibrosis, reinforcing the idea that PCM may trigger an autonomous fibrotic process leading to PH.

A retrospective clinical analysis of 510 patients with clinically cured PCM further substantiated these findings. Despite the absence of routine PH screening in PCM, 16 patients underwent echocardiographic evaluation, revealing that 8 exhibited RVSP values exceeding 35 mmHg, a threshold highly suggestive of PH. While most of these individuals were heavy smokers and had comorbid chronic obstructive pulmonary disease (COPD), which itself can contribute to vascular remodeling, PH in COPD typically remains mild, with the RVSP rarely exceeding 30 mmHg. The significantly elevated RVSP in PCM patients thus suggests that PCM acts as an independent driver of PH, rather than merely exacerbating preexisting pulmonary conditions. Collectively, these findings suggest that PCM induces a sustained proinflammatory and profibrotic microenvironment in the pulmonary vasculature, leading to progressive PH even after the pathogen’s clearance [82].

The observed adventitial myofibroblastic activation and increased perivascular collagen deposition provide a mechanistic insight into how PCM may contribute to the pathogenesis of PH, independent of other pulmonary comorbidities. Given the current lack of clinical guidelines recommending PH screening in PCM survivors, these results highlight the need for long-term cardiopulmonary follow-up in this patient population. The study underscores the necessity for further investigation to elucidate the full spectrum of PCM-induced pulmonary vascular remodeling and to determine whether targeted interventions could mitigate the progression of PH in these patients.

## 6. Participation of NETs in the Development of Lung Remodeling and Hypertension

In relation to NETs, these structures have been identified in the lesions of patients with PCM [38], along with the discovery of an important escape mechanism used by the fungus to evade NET-mediated action [35].

NETs play a crucial role in the activation of lung fibroblasts and their differentiation into myofibroblasts [73,74,75,83], and they are also involved in lungs’ epithelial injury and fibrosis [76,84]. Increasing evidence suggests that NETs may directly activate fibroblasts, while the inflammatory response triggered by NETs contributes to lung tissue damage, leading to secondary fibrosis [40,73].

Extracellular trap components, such as MPO and histones, have been shown to directly activate fibroblasts [40,77]. Neutrophil elastase (NE), a major protein component of NETs, has been implicated in the progression of pulmonary fibrosis, as its inhibition in murine models attenuates fibrosis by suppressing TGF-β1 signaling and inflammatory cell recruitment to the lungs [73,85]. Moreover, NETs likely activate the TGF-β signaling pathway in lung fibroblasts, further promoting pulmonary fibrosis [73].

These findings support the hypothesis that NETs could contribute to tissue remodeling, potentially leading to the pulmonary hypertension observed in PCM patients.

## 7. Conclusions

Emerging evidence highlights the significant role of fungal infections in the pathogenesis of pulmonary hypertension, largely driven by NET-induced mechanisms. Integrating echocardiographic screening for pulmonary hypertension in both early and late post-infection phases should be encouraged in clinical practice. In particular, immunocompromised patients with chronic or recurrent pulmonary fungal infections, especially those presenting with unexplained dyspnea or radiological evidence of vascular remodeling, may benefit from routine transthoracic echocardiography, as fatal outcomes due to undiagnosed pulmonary hypertension have already been reported in this population. This strategy may aid in the early identification of pulmonary hypertension as a contributing factor to increased morbidity and mortality, supporting the development of patient cohorts for experimental and clinical studies aimed at understanding the role of NETs in pulmonary hypertension.

## Figures and Tables

**Figure 1 idr-17-00084-f001:**
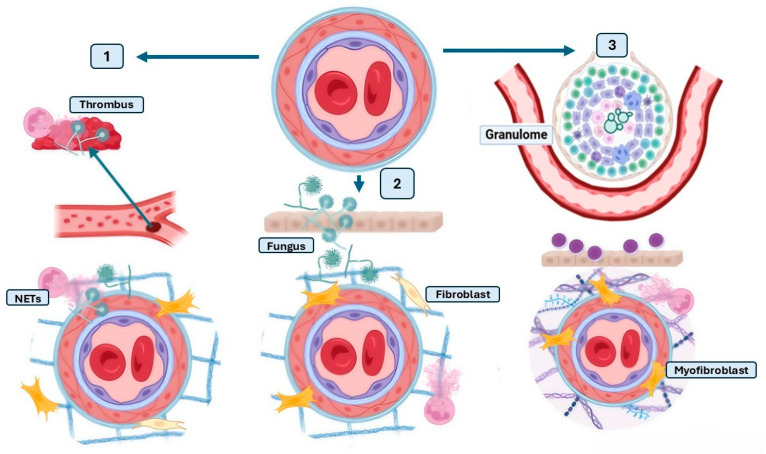
Pathophysiological panel of fungal infection-induced vascular remodeling associated with pulmonary hypertension. Vascular remodeling can occur in response to fungal infections and contribute to the development of pulmonary hypertension. This figure illustrates three key mechanisms: (1) Vascular remodeling after endothelial injury, in which endothelial disruption favors the formation of a thrombus and vascular inflammation. (2) Direct vascular remodeling during infection, characterized by neutrophil infiltration, the formation of neutrophil extracellular traps (NETs), and the activation of myofibroblasts and lymphocytes, promoting fibrosis and structural alterations in the blood vessel. (3) Post-infection distant vascular remodeling, where the formation of granulomas, in response to pathogens such as *Paracoccidioides* spp., stimulates the production of collagen, proteoglycans, and fibroblasts, affecting vascular homeostasis. Arrows indicate the progression from normal vasculature to infection-induced vascular remodeling. Created with Biorender.com (Available online: https://www.biorender.com/ (accessed on 2 July 2025)).

**Figure 2 idr-17-00084-f002:**
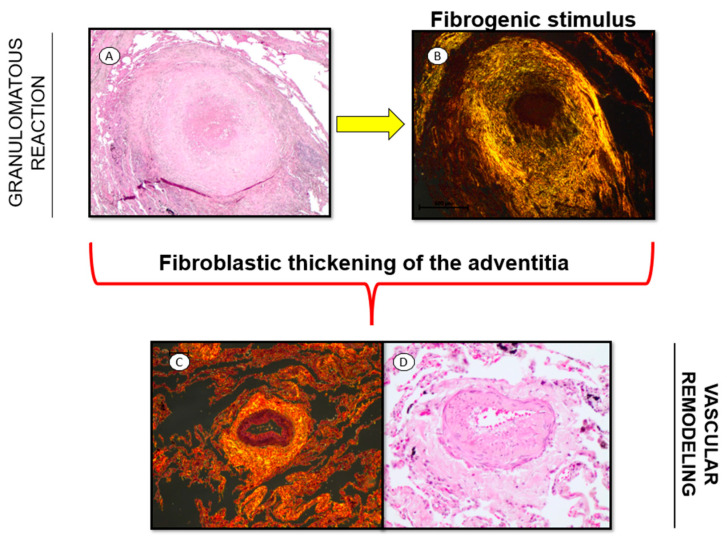
Histological panel from granulomatous inflammation to vascular remodeling. (**A**) Representative image of a well-circumscribed pulmonary granuloma exhibiting the typical histoarchitectural features of granulomatous inflammation, including a central zone of necrosis, as seen in H&E staining (200×), consistent with an organized immune response to a persistent fungal infection. (**B**) Prominent collagen deposition surrounding the granuloma, highlighted by Picrosirius Red staining (200×), indicates a fibrogenic microenvironment driven by myofibroblasts’ activation and the accumulation of extracellular matrix. (**C**) Collagen deposition is also evident within the vessel wall, as demonstrated by Picrosirius Red staining (200×), suggesting active perivascular fibrosis in association with chronic granulomatous inflammation. (**D**) Marked fibroblastic thickening of the vascular adventitia is observed on H&E staining (400×), accompanied by persistent structural distortion, characterizing ongoing vascular remodeling. The yellow arrow highlights that panels A and B correspond to the same lesion under different staining.

**Table 1 idr-17-00084-t001:** Pulmonary and cardiac findings associated with fungal Infections in humans and animal models.

Study Type	Agent	Lung Findings	Heart Findings	References
Autopsy study (n = 5)	*P. brasiliensis*	Intravascular thrombosis, granulomas, fibrosis, and emphysema.	Cor pulmonale	Yepez et al. [78]
Case report	*P. brasiliensis*	Pulmonary hypertension	-----------	Machado-filho et al. [79]
Clinical study (n = 14)	*P. brasiliensis*	Pulmonary hypertension	Cor pulmonale	Campos et al. [80]
Case report (n = 2)	*Aspergillus* sp.	Pulmonary artery hypertension	Endocarditis	Navabi et al., [8]
Case report	*Aspergillus fumigatus*	Pulmonary embolism Pulmonary hypertension	Endocarditis	Miranda et al. [9]
Case report	*Aspergillus* sp.	Pulmonary hypertension	Cor pulmonale	Bongomin et al. [23]
Case report	*P. lutzi*	Pulmonary hypertension	----	Gaspar et al. [81]
Clinical study and animal model	*P. brasiliensis*	Granulomas Pulmonary hypertension	-----	Batah et al. [82]
Case report	*Aspergillus fumigatus* and *Cryptococcus neoformans*	---------------	Acute heart failure	Ren et al. [22]
Animal model	*Aspergillus fumigatus*	Pulmonary arterial remodeling	------	Shreiner et al. [24]
Animal model	*Pneumocystis* sp.	Perivascular inflammation, medial hypertrophy, perivascular fibrosis, pulmonary hypertension	Right ventricular hypertrophy (RVH), elevated right ventricular pressure	Swain et al. [26,27]
Animal model	*Aspergillus fumigatus*	Interstitial pneumonia, pulmonary consolidation, fibrosis, vascular obstruction, pulmonary hypertension	Right ventricular hypertrophy and dilation, valvular insufficiency, ascites	Julian, Goryo [25]

## Data Availability

Data are available from the corresponding author upon request.

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
