# Peer review of "Pulmonary Hypertension Secondary to Fungal Infections: Underexplored Pathological Links"

_2036-7449, 2025, doi:10.3390/idr17040084_

Round 1
Reviewer 1 Report
Comments and Suggestions for Authors
In their mini-review the authors explore possible main mechanisms of vascular remodeling due to fungal infections leading to pulmonary hypertension.
The main topic of the manuscript is of interest for the reader, the manuscript is well structured and the figure supports the structure of the text. However, I would like to raise a few points to improve the manuscript:
- Overall, the manuscript is very therorethical and would improve by adding more data from humans with pulmonary hypertension after invasive fungal infection if available.
- Line 37: „eventually leading to vascular remodeling“: please refine how often this takes place in patients suffering from invasive fungal infections. Are there case reports possibly discussing one of the three mentioned key mechanisms? Based on this try to figure out how (clinically) relevant pulmonary hypertension secondary to fungal infections is. Is their published data on autopsies conducted in patients after invasive fungal infection and the prevalence of pulmonary hypertension?
- Which of the three mechanisms is considered to have the most impact on the development of pulmonary hypertension?
- Line 96: delete „antibiotic“ in „antibiotic treatment“ as this is correct treatment but maybe misleading in terms of treating a fungus with antibiotics
- Lines 100 following: please intalicize fungal names
- Line 105: please add full stop after sentence
- Please state at the beginning that your manuscript mainly focus on mechanisms leading to chronic pulmonary hyptension. Does (long-term) mechanical ventilation for example in patients with ARDS and concurrent fungal infection have an impact on development of chronic pulmonary hypertension and how can the different etologies be separated from each other? Please discuss this.
- Is there a link between (post)COVID-19-associated pulmonary aspergillosis and pulmonary hypertension? Please discuss this in more detail.
- Line 118: the abbreviation PPCM is used for the first time for „patients with paracoccidioidomycosis“ but further on is used as abbreviation just for paracoccidioidomycosis. Please change the abbreviation „PPCM“ to „PCM“ in the consecutive text
- Lines 122/123: please delete first names of authors in the in-text citations
- Paragraph 4: Possibly due to the brazilian affiliation of the authors, the main focus of this paragraph is on Paracoccidioides. Also, most literature about involvement of fungi in the development of pulmonary hypertension is about Paracoccidioides. Compared to the other paragraphs, it is more detailed and includes data on humans which is preferable. This should be adopted for the other paragraphs if possible. However, just one reference is given in the whole paragraph. Please revise.
Author Response
"Please see the attachment"

Reviewer 2 Report
Comments and Suggestions for Authors
Rodríguez-Herrera's study addresses a truly interesting and underexplored topic: pulmonary hypertension secondary to fungal infection. Three mechanisms underlying vascular remodeling caused by pulmonary hypertension are addressed. The work is presented in an easily understandable manner, and the conclusions are based on the data found in the literature they reviewed.
My main criticism is about the study design, as it is a narrative review, which, compared to a systematic review, is less objective, transparent, and methodologically rigorous. There are articles that address the topic of hypertension associated with fungal infection, for example Swain, S. D., Han, S., Harmsen, A., Shampeny, K., & Harmsen, A. G. (2007). Pulmonary hypertension can be a sequela of prior Pneumocystis pneumonia. The American journal of pathology, 171(3), 790–799. https://doi.org/10.2353/ajpath.2007.070178, but are not included in this review. This is an important limitation that the authors should state in the manuscript, as it leads to information bias.
Minor corrections
“Review” is not a keyword; please remove it.
Cite according to the IDR guidelines for authors.
Lines 39-40 briefly explain why evaluation for pulmonary hypertension in patients with fungal infections is not recommended. Include references.
Figure 1. Place the numbers (1), (2), and (3) as appropriate on the image. Write spp. without italics.
Lines 100-103: Write scientific names in italics.
Line 118: Change (PPCM) to (PCM). Correct throughout.
Lines 122 and 127: Change Paracoccidioides brasiliensis to P. brasiliensis.
Line 123: Change Paracoccidioides lutzii to P. lutzii.
Lines 134-135: Include the corresponding reference.
Author Response
"Please see the attachment"

Reviewer 3 Report
Comments and Suggestions for Authors
Rodríguez-Herrera and 5 colleagues submit a manuscript regarding underexplored pathological links to pulmonary hypertension secondary to fungal infections.
Comments:
- Since this is a paper about pathology, please include black and white or color photographs of actual histology from human tissues that could complement some of the theories described in the paper.
- What is interstitial lung Tumor disease (ILD) and how is it different from the more known interstitial lung disease (ILD)?
- Section 4, Underlying mechanisms, is a single paragraph. Please consider rewriting and breaking this up into several paragraphs.
- Latin genus and species names should be in italics.
- Reference 1 has an author formatting issue. Please correct.
- Page numbers are missing from reference 2, 3, 9, 10, 18, 25, 27, 30, 31, 32, 33, 35, 39, 40, 46, 47, 48, 49, 50, 51, 52, 53, 54, 55, 56, 58, 60, 61, 62, 63, 66, 67, 68, 73, 77, 78. Please add.
Author Response
"Please see the attachment"

Reviewer 4 Report
Comments and Suggestions for Authors
This mini-review addresses a timely and often overlooked subject - the potential link between pulmonary fungal infections and pulmonary hypertension (PH). The authors present a clear structure, focusing on three key pathways of vascular remodeling: following endothelial injury, during ongoing infection and after infection resolution. The discussion around neutrophil extracellular traps (NETs) as a shared mechanism driving inflammation and fibrosis adds meaningful depth to the review. The manuscript is well-supported by both experimental and clinical literature, especially in relation to paracoccidioidomycosis (PCM). That said, several aspects could be improved to strengthen the overall impact of the work.
Major Points for Improvement:
- The introduction would benefit from a more clearly defined central hypothesis. As it stands, it summarizes existing knowledge without clearly stating the core research question or rationale.
- The manuscript doesn’t always differentiate between strong, direct evidence and more speculative or associative data. Clarifying the type and strength of evidence, whether it’s based on clinical observations, animal models, or in vitro studies that would help readers better interpret the findings.
- While the conclusion mentions the potential value of PH screening in affected patients, a more practical discussion on how this could be integrated into clinical care, especially for immunocompromised individuals, would greatly enhance the review’s applicability.
Suggestions for Enhancement:
- Consider including a table summarizing key fungal pathogens, their reported associations with PH, and the type of supporting evidence (e.g., clinical cases, experimental models).
- Figure 1 is informative but could be improved for clarity like simplifying the visual elements and streamlining the legend would help make the message more accessible.
Author Response
"Please see the attachment"

Round 2
Reviewer 2 Report
Comments and Suggestions for Authors
The manuscript has improved, thanks for listening to the suggestions.
Author Response
Comment: The manuscript has improved, thanks for listening to the suggestions.
Response: We sincerely appreciate your positive feedback. We are glad the revisions have improved the manuscript
Reviewer 3 Report
Comments and Suggestions for Authors
Rodríguez-Herrera have submitted a revised version of their manuscript. It is improved. There are still some small items that could be improved.
Comments:
- The term "spp.", the species designation, does not need to be in italics when used.
- Some references are not complete. See lack of page numbers for reference 3, 29, 33, 34, 40, 44, 45, 51, 52, 53, 54, 55, 56, 57, 58, 59, 60, 61, 63, 65, 66, 67, 68, 71, 72, 73, 78, 82.
- Page numbering for the following references might be incorrect, so please double check: 11, 22.
- References 12 and 13 have the exact same page numbers. Please check to make sure that these are correct.
Author Response
Comment1: The term "spp.", the species designation, does not need to be in italics when used.
Response1: Thanks for you suggestion, we made the corrections in line 70 and 146.
Comment 2: Some references are not complete. See lack of page numbers for reference 3, 29, 33, 34, 40, 44, 45, 51, 52, 53, 54, 55, 56, 57, 58, 59, 60, 61, 63, 65, 66, 67, 68, 71, 72, 73, 78, 82.
Response 2: Thanks for the observation, we added the page numbers like the example:
Zuo Y, Yalavarthi S, Shi H, Gockman K, Zuo M, Madison JA, et al. Neutrophil extracellular traps in COVID-19. JCI Insight. 2020, 5, e138999.
Reusch N, De Domenico E, Bonaguro L, Schulte-Schrepping J, Baßler K, Schultze JL, et al. Neutrophils in COVID-19. Frontiers in Immunology 2021, 12, 652470.
Arcanjo A, Logullo J, Menezes CCB, de Souza Carvalho Giangiarulo TC, dos Reis MC, de Castro GMM, et al. The emerging role of neutrophil extracellular traps in severe acute respiratory syndrome coronavirus 2 (COVID-19). Scientific Reports. 2020, 10, 19630.
Barnes BJ, Adrover JM, Baxter-Stoltzfus A, Borczuk A, Cools-Lartigue J, Crawford JM, et al. Targeting potential drivers of COVID-19: Neutrophil extracellular traps. Journal of Experimental Medicine 2020, 217, e20200652.
Middleton EA, He XY, Denorme F, Campbell RA, Ng D, Salvatore SP, et al. Neutrophil extracellular traps contribute to immunothrombosis in COVID-19 acute respiratory distress syndrome. Blood. 2020, 136(10), 1169–79.
Al-Kuraishy HM, Al-Gareeb AI, Al-hussaniy HA, Al-Harcan NAH, Alexiou A, Batiha GES. Neutrophil Extracellular Traps (NETs) and Covid-19: A new frontiers for therapeutic modality. International Immunopharmacology, 2022, 104, 108516.
Nicolai L, Leunig A, Brambs S, Kaiser R, Weinberger T, Weigand M, et al. Immunothrombotic Dysregulation in COVID-19 Pneumonia Is Associated With Respiratory Failure and Coagulopathy. Circulation. 2020, 42(12), 1176–89.
Veras FP, Pontelli MC, Silva CM, Toller-Kawahisa JE, de Lima M, Nascimento DC, et al. SARS-CoV-2-triggered neutrophil extracellular traps mediate COVID-19 pathology. Journal of Experimental Medicine 2020, 217(12), 20201129.
Weber AG, Chau AS, Egeblad M, Barnes BJ, Janowitz T. Nebulized in-line endotracheal dornase alfa and albuterol administered to mechanically ventilated COVID-19 patients: A case series. Molecular Medicine 2020, 26, 1-7.
Toma A, Darwish C, Taylor M, Harlacher J, Darwish R. The Use of Dornase Alfa in the Management of COVID-19-Associated Adult Respiratory Distress Syndrome. Crit Care Res Pract. 2021, 2021, 8881115.
Shao BZ, Yao Y, Li JP, Chai NL, Linghu EQ. The Role of Neutrophil Extracellular Traps in Cancer. Frontiers in Oncology 2021, 11, 714357.
Comment 3: Page numbering for the following references might be incorrect, so please double check: 11, 22.
Response 3: Thanks for you comment, we corrected the numbers.
- Earle K; Valero C; Conn DP; Vere G; Cook PC; Bromley MJ; et al. Pathogenicity and virulence of Aspergillus fumigatus. Virulence 2023, 14, 2172264.
- Ren H; Zhao Q; Jiang J; Yang W; Fu A; Ge Y. Acute heart failure due to pulmonary Aspergillus fumigatus and Cryptococcus neoformans infection associated with COVID-19. Lab. 2023, 69, 2011-2016.
Comment 4: References 12 and 13 have the exact same page numbers. Please check to make sure that these are correct.
Response 4: we check the number and correction was made. See line 291.
Reviewer 4 Report
Comments and Suggestions for Authors
The authors have thoughtfully and thoroughly addressed all the reviewer comments. The introduction now clearly outlines the central hypothesis, setting a strong foundation for the manuscript. Throughout the text, the distinctions between clinical findings, experimental data, and more speculative associations are now much clearer. The revised conclusion offers practical suggestions for incorporating pulmonary hypertension screening into clinical practice, especially for immunocompromised individuals. The addition of a well-organized summary table (Table 1) effectively highlights key fungal pathogens and the type of evidence supporting their link to PH. Finally, Figure 1 has been streamlined for better readability and impact.
Author Response
Response: Thank you for your kind words. We are pleased to know that the changes addressed your suggestions effectively.